# Investigation of the Biosafety of Antibacterial Mg(OH)_2_ Nanoparticles to a Normal Biological System

**DOI:** 10.3390/jfb14040229

**Published:** 2023-04-18

**Authors:** Ying Wang, Yanjing Liu, Xiyue Li, Fuming Wang, Yaping Huang, Yuezhou Liu, Yimin Zhu

**Affiliations:** 1School of Textile and Material Engineering, Dalian Polytechnic University, Dalian 116034, China; 2Collaborative Innovation Central for Vessel Pollution Monitoring and Control, Dalian Maritime University, Dalian 116026, China

**Keywords:** biosafety, Mg(OH)_2_ NPs, toxicity, irritation, cytotoxicity

## Abstract

The toxicity of Mg(OH)_2_ nanoparticles (NPs) as antibacterial agents to a normal biological system is unclear, so it is necessary to evaluate their potential toxic effect for safe use. In this work, the administration of these antibacterial agents did not induce pulmonary interstitial fibrosis as no significant effect on the proliferation of HELF cells was observed in vitro. Additionally, Mg(OH)_2_ NPs caused no inhibition of the proliferation of PC-12 cells, indicating that the brain’s nervous system was not affected by Mg(OH)_2_ NPs. The acute oral toxicity test showed that the Mg(OH)_2_ NPs at 10,000 mg/kg induced no mortality during the administration period, and there was little toxicity in vital organs according to a histological analysis. In addition, the in vivo acute eye irritation test results showed little acute irritation of the eye caused by Mg(OH)_2_ NPs. Thus, Mg(OH)_2_ NPs exhibited great biosafety to a normal biological system, which was critical for human health and environmental protection.

## 1. Introduction

Antibacterial nanomaterials have been designed to inhibit the growth of bacteria and control the spread of infectious disease [1,2,3,4], and have been used in a wide range of fields such as textile, ceramic and plastic [5,6]. As nanotechnology rapidly expands, antibacterial nanoparticles (NPs) are entering the environment, and it is inevitable that human will be exposed to the NPs [7,8,9]. Thus, the biosafety of antibacterial NPs has been increasingly evaluated [10,11].

Mg(OH)_2_ NPs as attractive antibacterial materials have been found to have excellent capability to kill bacteria, and their utilization in various fields such as clothing, cosmetic, electronics and medicine has increased exponentially [12]. Mg(OH)_2_ NPs combining both nanotechnology and antibacterial technology are known for their small size, easy synthesis and long-term stability [13,14,15]. Additionally, Mg(OH)_2_ NPs can exert their antibacterial effect in the absence of light [16]. The antibacterial mechanism of Mg(OH)_2_ NPs is commonly attributed to the production of reactive oxygen species (ROS) on the Mg(OH)_2_ NP surface, which can cause bacterial lipid peroxidation and death [17,18,19]. However, the potential toxicity of Mg(OH)_2_ NPs to a normal biological system is still unclear. Although the ROS is the potential antibacterial mediator of Mg(OH)_2_ NPs, it is necessary to ensure that the level of ROS is tolerable to and safe for a normal biological system. Thus, as a critical prerequisite prior to market introduction, the biosafety of Mg(OH)_2_ NPs is yet to be examined.

Due to their small size, Mg(OH)_2_ NPs can enter the human body via the mouth, eye and nose. A toxicokinetic study showed that a significant level of Mg accumulates in liver and kidney tissues but not in the urine and feces [20]. Thus, intentional or accidental exposure to Mg(OH)_2_ NPs may lead to an accumulation of Mg in the human body. Recently, an investigation reported that Mg(OH)_2_ NPs exhibit no significant genotoxic effect on human liver epithelial cancer cells at the concentrations of 25, 75 and 150 µg/mL, implicating their potential applications in nanomedicine [21]. In addition, Mg(OH)_2_ NPs have less of a cytotoxic effect on human cardiac microvascular endothelial cells than of ZnO NPs and CuO NPs do [22]. It has also been reported that low-concentration Mg(OH)_2_ NPs have no cytotoxicity to human umbilical vein endothelial cells, which can enhance NO release and the total antioxidation competence of the T-AOC content [23]. Moreover, the total antioxidant capacity of the serum of rats was reduced after the instillation of Mg(OH)_2_ NPs in cases of acute intratracheal toxicity [24]. Thus, the accumulation of Mg in the human body is due to the extensive applications of Mg(OH)_2_ NPs, and the biosafety evaluation is imperative considering the promotion of Mg(OH)_2_ NP products and the protection of human health and the environment [25].

In this work, synthetic Mg(OH)_2_ NPs were characterized by scanning electron microscopy (SEM), X-ray diffraction (XRD), energy-dispersive X-ray spectrometry (EDS) and X-ray photoelectron spectroscopy (XPS). The safe dose was evaluated by the acute oral toxicity test. The effect on vital organs was determined by the histopathological analysis. The irritation to the eye was evaluated by the acute irritation test in vivo. Additionally, the cytotoxicity effect of Mg(OH)_2_ NPs on respiratory and brain nervous systems was evaluated through the in vitro test of human embryonic lung fibroblast (HELF) cells and rat adrenergic neural phaeochromocytoma tumor (PC-12) cells. The aforementioned experiments comprehensively evaluated the biosafety of Mg(OH)_2_ NPs to a normal biological system.

## 2. Materials and Methods

### 2.1. Materials

The analytical-grade chemicals without further purification were purchased from Sigma-Aldrich (St. Louis, MO, USA). Deionized water was purified by an Elgastat Spectrum reverse osmosis system (Elga LTD, High Wycombe, UK). Kun Ming (KM) mice (211002300021330131) and Japanese white (JW) rabbits (211001600000693) were purchased from Shenyang Changsheng Biotechnology Co., Ltd. (Shenyang, China) (SCXK Liao 2015-0001) and maintained in a thermostatic laboratory. In addition, human embryonic lung fibroblast (HELF) cells (MRC-5 system) and rat adrenergic neural phaeochromocytoma tumor (PC-12) cells were obtained from Shenyang Pharmaceutical University. All the experiments were performed in compliance with the animal management rules of the Health Ministry of China and the Shenyang Pharmaceutical University’s committee on the use and care of laboratory animals.

### 2.2. Synthesis of Mg(OH)_2_ NPs

The Mg(OH)_2_ NPs were prepared from the co-precipitation of MgCl_2_·6H_2_O, PEG and NH_3_·H_2_O at 60 °C for 1.5 h [26,27]. An amount of 200 mL of the MgCl_2_ solutions (1.5 mol/L) were added to a 500 mL three-mouth flask, 0.4 g PEG was added as dispersant, and the three-mouth flask was placed in an ultrasonic bath to accelerate the dissolution of PEG. Then, 5 wt% NH_3_·H_2_O was added slowly dropwise into the mixture (n(OH^−^)/n(Mg^2+^) ≥ 2). The mixture was stirred at 300 rpm at 40 °C for 1.5 h. The resulting suspension was cooled to room temperature (20 °C) and aged for 12 h to allow the complete precipitation of Mg^2+^. The resulting suspension was filtered and washed with ionized water and anhydrous ethanol, respectively. It was dried under a vacuum at 80 °C for 4 h and ground uniformly to obtain the product nano-Mg(OH)_2_ [26,27].

### 2.3. Characterization

The X-ray diffraction (XRD, Rigaku D/max-2500/PC) was used to analyze the crystal structure and purity of the Mg(OH)_2_ NPs using Cu Kα radiation (λ = 0.15418) at 25 mA and 40 kV, which was acquired from 5° to 90° with a step size of 0.05°/s. A scanning electron microscope (SEM, NOVA Nano SEM 450) was operated at the accelerating voltage of 3–20 kV to analyze the surface morphology of the Mg(OH)_2_ NPs. In addition, the conductive films with gold were coated on the Mg(OH)_2_ NP surfaces for the SEM test. The elemental composition of the sample was tested by the scanning electron microscope (EDS, X-Max50, Oxford Instruments, Abingdon, UK) and X-ray photoelectron spectroscopy (XPS, Thermo ESCALAB 250Xi, Al K-Alpha) on a VG MultiLab 2000 X-ray photoelectron spectrometer (Thermo Electron Corporation, Waltham, MA, USA) using Al-Kα (hλ = 1486.6 eV) radiation as the excitation source, and the spectra were calibrated by the C 1s peak (284.8 eV).

### 2.4. Culture and Treatment of HELF Cells and PC-12 Cells

The HELF cells and PC-12 cells were maintained in Dulbecco’s modified eagle medium (DMEM) containing 10% fetal bovine serum as single-cell suspensions after centrifugation at 1000 rpm and cultured at 37 °C [28] in a 5% CO_2_ humidified incubator. The Mg(OH)_2_ NPs were added to DMEM to obtain a range of concentrations (0, 0.5, 1.0, 1.5, 2.0, 2.5, 3.0, 3.5, 4.0, 4.5, and 5.0 mg/mL), as shown in Figure 1. In addition, 5 × 10^3^ cells per well were inoculated into 96-well plates(in an amount of 100 µL for 3 generations and 24 h). After the cells were fully attached, the original medium was replaced with 100 µL of the Mg(OH)_2_ NP suspensions or 100 µL of the normal complete culture media for the control group. After 72 h of incubation in the incubator, cell viability was determined by the 3-(4,5-dimethylthiazole)-2,5 diphenyltetrazolium bromide (MTT) method by adding 15 µL of a 5 mg/mL MTT working solution per well and continuing incubation in the incubator for 4 h. Subsequently, 150 μL of DMSO was added to each well and shaken for 10 min, cell viability was calculated by Equation (1), and the optical densitometry (OD) values at 570 nm were measured using an enzyme plate tester (SpectraMax i3x). In addition, the morphology of the cells was observed under an inverted microscope (KOSTER IMC 800Ti).
(1)$$CV=\frac{ODe}{ODc}\times 100\%$$
where *CV* is the cell viability; *ODe* is the OD value of the experimental group; *ODc* is the OD value of the control group.

### 2.5. Acute Oral Toxicity Test and Histopathological Sections

The acute oral toxicity experiment was performed in a constant-temperature animal laboratory (25 °C), and the KM mice were kept for 3 d to adapt to the laboratory environment, while the growth and disease statuses of KM mice were observed (GBZ/T 240.2-2011). As shown in Figure 1, 20 healthy KM mice without abnormal behavioral signs were divided into two groups and fasted for 16 h. The control group was given sterile water (the solvent was sterile deionized water), and the experimental group was given a suspension of 10,000 mg/kg of Mg(OH)_2_ NPs in sterile water as a solvent for the one-time oral gavage of KM mice. After administration, the mice were normally housed and observed for abnormalities in skin condition, hair volume, respiratory rate and eye secretions at 0, 4, 7, 11 and 14 d, especially for clinical signs of toxicity such as comas and drowsiness. Mortality was recorded after 14 d of treatment in accordance with the methods of some researches [26,27].

After the acute oral toxicity test, the KM mice were executed by the cervical dislocation method and analyzed for histopathological changes after drug administration. The excised organs were fixed in 4% paraformaldehyde, decalcified in EDTA, and then dehydrated in a series of graded ethanol (70%, 80%, 95% and 100%) and embedded in paraffin to prepare tissue sections. Longitudinal sections of organs (5 µm thick) were stained with hematoxylin-eosin (HE) and by Giemsa staining, and pathological changes in organ tissue were observed with transmission light microscopy (CKX41) [29].

### 2.6. In Vivo Acute Eye Irritation Test

The Draize method was used for the acute eye irritation toxicity experiment. The animals had to be assessed at least twice daily, for the first three days after the administration of the Mg(OH)_2_ NPs. The JW rabbits were kept in a constant-temperature animal laboratory for 3 d before the experiment to adapt to the laboratory environment, while the growth and disease statuses were observed. As shown in Figure 1, the eyes of four healthy, disease-free JW rabbits were selected and kept for 3 d. Then, both eyes of each JW rabbit were examined to confirm the absence of signs of eye irritation (conjunctival rupture, corneal defects and iris damage). The JW rabbits were fixed, the left lower eyelid was gently pulled down, the Mg(OH)_2_ NPs (0.1 g) were immediately added into the conjunctival sac and the eyes were closed for 1 min. The untreated right eyes were used as controls. After 0, 1, 24, 48, and 72 h of administration, the conjunctiva, cornea, and iris of the JW rabbit eyes were examined with a magnification glass and assessed for abnormalities, which were recorded. Table 1 shows the scoring criteria for acute ocular irritation. There was no animal discomfort sign including repeated pawing or rubbing of the eye, excessive blinking or excessive tearing during this process.

## 3. Results

### 3.1. Characterizations of Mg(OH)_2_ NPs

Due to the fact that the toxic effect of inorganic NPs depends on their size and shapes, the micromorphology of Mg(OH)_2_ NPs is characterized through SEM and XRD. The SEM image of the Mg(OH)_2_ NPs in Figure 2a show that the particle size was 30~50 nm. The XRD pattern in Figure 2b shows that the lattice constants were comparable to the values of the Joint Committee on Powder Diffraction Standards (JCPDS 07-0239), and all the diffraction peaks were well-indexed as a Mg(OH)_2_ structure. In addition, the purity was very high as no impure peaks were observed other than the characteristic peaks of Mg(OH)_2_. Based on the Scherrer equation, Equation (2), the average particle size of the Mg(OH)_2_ NPs was 35.9 nm, which was consistent with the SEM result.
(2)$$D=\frac{K\cdot \lambda }{B\cdot \cos\theta }$$
where *D* is the particle size (nm); *K* is the Scherrer constant (0.89); λ (nm) is the diffraction wavelength (0.15418 Å); *B* is the half width of the diffraction peak; *θ* is the diffraction angle.

The composition of the surface elements of the synthesized Mg(OH)_2_ NP products was studied by an EDS spectrometer, and the EDS values representing the strong characteristic peaks of magnesium and oxygen elements were shown. No other impurity elements were observed in the spectrum, indicating the high purity of the nanoparticle structure. The surface scanning of elements C, O, N and Mg is shown in Figure 3a. The elements are presented with different colors. The atomic percentage ratio of Mg to O elements was 27.41:56.69, and the atomic number ratio of Mg to O elements was close to 1:2. The weight percentages of the Mg and O in the product were 42.34 and 52.32, respectively, which are consistent with the theoretical values. The EDS spectra showed that the synthesized product had Mg(OH)_2_ NPs.

XPS was used to analyze the surface chemical composition of the Mg(OH)_2_ NPs. Figure 3b is the full-scan spectrum of the sample, which shows that the test sample mainly contained the characteristic peaks of Mg, O and C. No miscellaneous peaks of other substances appeared, as shown in Figure 3b, indicating the high purity of the product. The XPS atlas shows that the characteristic peak of Mg2p appeared at 49.6 eV and that that of Mg1s appeared at 1303.14 eV. In the sample, Mg2p fitted only one peak, which should be attributed to the Mg^2+^ in Mg(OH)_2_. The characteristic peak of O1s appeared at 531.12 eV, which can also indicate the high purity of the generated Mg(OH)_2_ NPs.

### 3.2. In Vitro Cytotoxicity of Mg(OH)_2_ NPs to HELF Cells and PC-12 Cells

As shown in Figure 2c, the cell viability of HELF cells showed no obvious fluctuation after the cells were treated with different concentrations (0, 0.5, 1.0, 1.5, 2.0, 2.5, 3.0, 3.5, 4.0, 4.5, and 5.0 mg/mL) of Mg(OH)_2_ NPs, which indicated that the Mg(OH)_2_ NPs had no significant toxic effect on the proliferation of the HELF cells. Additionally, the morphology of the HELF cells shown in Figure 2e remained normal after the treatment with different concentrations of Mg(OH)_2_ NPs.

Figure 2f confirms the cell viability of PC-12 cells and shows that there was no significant inhibition of PC-12 cells with the concentrations of Mg(OH)_2_ NPs ranging from 0 mg/mL to 5.0 mg/mL. In addition, the PC-12 cell morphology shown in Figure 2d was similar to that of the control group. Therefore, the cytotoxicity result indicated that Mg(OH)_2_ NPs had no cytotoxicity to HELF cells and PC-12 cells.

### 3.3. Acute Oral Toxicity and Histopathology Research

The acute oral toxicity test can allow one to find out the lethal dose of the tested chemical, and the preliminary evaluation of the toxic effect characteristics, target organs, dose–response relationship and the risk of damage to the human body posed by nano-Mg(OH)_2_ by observing the toxic effect performance, intensity of the toxic effect and the death of animals, so the method is reproducible and can be used for acute toxicity evaluation with fewer animals. The skin, hair, respiratory and eye secretion contents of the KM mice administered Mg(OH)_2_ NPs were the same as those of the control group. Additionally, after 14 d, there was no clinical toxicity indicators such as comas, diarrhea, lethargy, nausea, salivation, vomiting, convulsions or tremors, and the mortality ratio was 0%. Thus, the KM mice appeared to be normal with no sign of toxicity at a dose of 10,000 mg/kg administered orally, and the lethal dose of Mg(OH)_2_ NPs was found to potentially be more than 10,000 mg/kg, which fell into the actual non-toxic range.

The most direct way to diagnose pathological changes in organs is to prepare pathological tissue sections. The histological sections of KM mice after 14 d of transoral administration were analyzed, as shown in Figure 2g. The pathological sections of all organs in the experimental group showed no abnormal changes compared with those of the control group. In the heart tissue, there was no inflammatory infiltration between cardiac myocytes, no myocarditis, and no granulation, vacuolation or fatty degeneration. The brain tissue sections showed no demyelination and inflammatory infiltrates in the brain and no abnormalities in astrocytes [30]. There was no inflammatory cell infiltration in the lungs, no fibroplasia around the small airways, and no fibrosis in the interstitial lung. There were no signs of fibrosis in the myocardial fibers, much less degeneration or fragmentation, and fewer vitreous changes. There was no blockage of blood vessels, no capillary abnormalities, and no swelling and detachment of endothelial cells. There were no inflammatory infiltrates, no instances of interstitial inflammation, no glomerular lesions and no glomerular vascular tract cells, and there was no endothelial cell proliferation in the kidney tissues. The stomach tissue was intact with a normal morphology of mural and principal cells. No necrosis was observed in the liver tissue and no high concentration of leukocytes was seen in the connective tissue, which indicates that no inflammation was generated in the liver after transoral administration. The size of lymphoid follicles in the spleen tissue of the experimental group was normal and comparable to that of the control group. Pathological sections of the large intestinal organs showed a normal intestinal structure without the infiltration of lymphocytes and plasma cells in the lamina propria, the villi on the surface of the large intestine were not swollen and congested, and the intestinal glands did not bleed. In conclusion, the brain, heart, lung, kidney, stomach, liver and spleen of Mg(OH)_2_ NP-treated KM mice behaved normally without signs of necrosis or inflammation [31]. Therefore, Mg(OH)_2_ NPs were less toxic to the above vital organs after 14 d of administration.

### 3.4. In Vivo Acute Eye Irritation

The acute eye irritation toxicity test (Draize test) was performed by making the Mg(OH)_2_ NPs come into contact with the eyes of the JW rabbits, and the irritation level of magnesium hydroxide nanoparticles could be directly measured. Due to the sensitivity of the rabbit eye itself, the irritation effect of Mg(OH)_2_ NPs on the human eye could be predicted, and the potential hazard of Mg(OH)_2_ NPs to the human eye could be easily identified. The acute eye irritation score is shown in the bottom part of Figure 4 and in Table 2. There was no change in the control eyes and the experimental eyes after 0, 1, 24, 48 and 72 h. There was no opacity or damaged area in the corneas. The irises showed no abnormality and the pupils responded to light. In addition, there was no hyperemia or edema of the conjunctiva and the amount of eye secretions remained normal in the experimental group. The data showed that Mg(OH)_2_ NPs at 0.1 g induced no acute eye irritation in the in vivo model.

## 4. Discussion

It is necessary to assess the biosafety of Mg(OH)_2_ NPs to promote their extensive applications. The characterizations of Mg(OH)_2_ NPs were necessary to understand their possible toxicity. The XRD and SEM results showed that the high-purity Mg(OH)_2_ NPs in this study had a particle size range of 30~50 nm, which was less than 100 nm at the nanoscale (Figure 2e,f). It has been reported that NPs may invade vital organs easily due to their small particle size and thereby cause toxicity to the human body [32,33,34]. Hence, the aim of our study was to assess the possible toxicity of Mg(OH)_2_ NPs after acute oral administration, the in vivo acute irritation to eye, as well as the effect on respiratory and brain nervous systems. Thus, the possible toxicity of Mg(OH)_2_ NPs to a normal biological system was investigated comprehensively.

The respiratory system maintains normal life, while nanoscale particles can enter the alveolus through respiration. Toxic particles may not only damage the respiratory system, but also many other systems through blood circulation in the alveolar capillaries. Pulmonary interstitial fibrosis is a common and irreversible pulmonary disease, and the proliferation of HELF cells plays a catalytic role in it [35]. Toxicity-induced HELF cells transform into myofibroblasts and secrete a large amount of collagen. A large number of myofibroblasts and a large amount of collagen being accumulated in the lung may cause irreversible pulmonary interstitial fibrosis and eventually result in lung failure. The histological study showed that Mg(OH)_2_ NP-treated lungs had no necrosis, which was a similar case to the control group (Figure 2g). In this work, HELF cells as an in vitro model were chosen to imitate the environment of the respiratory tract, and the effect of Mg(OH)_2_ NPs on the proliferation of HELF cells was determined by MTT assays. The viability of HELF cells was not affected by the treatment, indicating that the Mg(OH)_2_ NPs at different concentrations had no effect on the proliferation of HELF cells (Figure 2e). The morphology of the HELF cells treated with Mg(OH)_2_ NPs was similar to that of the control group (Figure 2d). Thus, Mg(OH)_2_ NPs are unlikely to induce lung fibrosis and are safe for the respiratory system.

Invasive NPs can be distributed across the human brain via blood circulation, which may lead to brain nervous system disease. A lot of research has shown that ROS oxidative stress plays an important role in brain nervous disease [36]. However, the effect of Mg(OH)_2_ NPs on the human brain’s nervous system remains unknown. As shown in Figure 2g, dopaminergic neurons present in the human brain can regulate a variety of physiological and behavioral processes. When the level of these is reduced or abnormally delivered, the body is slow in response and action. PC-12 cells can secrete dopamine, are used extensively as an in vitro model for dopaminergic neurons and chosen to imitate the brain’s nervous system. The PC-12 cell viability indicated that Mg(OH)_2_ NPs in different concentrations were less likely to induce the death of PC-12 cells (Figure 2d). The morphology of the PC-12 cells treated with Mg(OH)_2_ NPs was similar to that of the control group (Figure 2f). The results indicate that Mg(OH)_2_ NPs have excellent biosafety to the brain nervous system. Of course, the in vitro culture conditions of HELF cells and PC-12 cells can be controlled, so that there are less factors influencing cell viability. 

Although the acute oral toxicity test is unable to reflect the effect of Mg(OH)_2_ NPs on the functions of respiratory and brain nervous systems, it is used to determine a lethal dose for safe use [37]. The acute oral toxicity test showed that the lethal dose of Mg(OH)_2_ NPs for safe use was more than 10,000 mg/kg and that the corresponding toxicity level was non-toxic. The histological images indicate that the organs of the experimental group were similar to those of the control group. The liver showed no necrosis or a large number of leukocytes in the connective tissue, suggesting that there was no inflammation in the liver after administration. The cardiac fiber in the heart tissue was normal without any degeneration, fragmentation and hyalinization. Additionally, there were no congested blood vessels, which was similar to the case of the control mice’s heart architecture. The spleen could be a direct or indirect target of toxicity. Our results show that there was no significant change in the spleen lymphatic follicle tissue upon treatment. Thus, Mg(OH)_2_ NPs had no effect on the vital organs based on the histological analysis (Figure 2g).

At present, antibacterial materials of Mg(OH)_2_ NPs are being widely used in daily life for commercial purposes [38]. However, little research is available on in vivo irritation to the eye. Thus, the evaluation of acute eye irritation is a significant aspect of the wide applications of Mg(OH)_2_ NPs. The results show that there was no visible edema or erythema in the cornea, iris and conjunctiva of the eye model compared to the control group (Table 2 and Figure 4). Thus, there was no acute irritation when Mg(OH)_2_ NPs made contact with the eye. However, the physiological environment of the respiratory system and brain nervous system in the normal body is complex, and is affected by many uncontrollable factors. Thus, it is necessary to conduct in vivo studies to confirm the theoretical basis for the applications of Mg(OH)_2_ NPs.

## 5. Conclusions

In summary, the biosafety of Mg(OH)_2_ NPs to a normal biological system was investigated comprehensively. An acute oral toxicity test showed that Mg(OH)_2_ NPs induced no toxicity or effects on the vital organs of KM mice, and the lethal dose for safe use was more than 10,000 mg/kg. There was no acute irritation in vivo when Mg(OH)_2_ NPs came into contact with the eye. In addition, the Mg(OH)_2_ NPs caused no pulmonary interstitial fibrosis in vitro. Additionally, the Mg(OH)_2_ NPs had no effect on dopaminergic neurons and exhibited excellent biosafety to the brain nervous system in vitro. Therefore, we conclude that Mg(OH)_2_ NPs can be safely used in a wide range of applications. However, caution should be taken while they are used in medical fields, considering any potential health hazards. Furthermore, the purpose of research on the biosafety of Mg(OH)_2_ NPs is not to restrict its applications, but to find solutions to overcome the problems that may arise and achieve its safe use eventually.

## Figures and Tables

**Figure 1 jfb-14-00229-f001:**
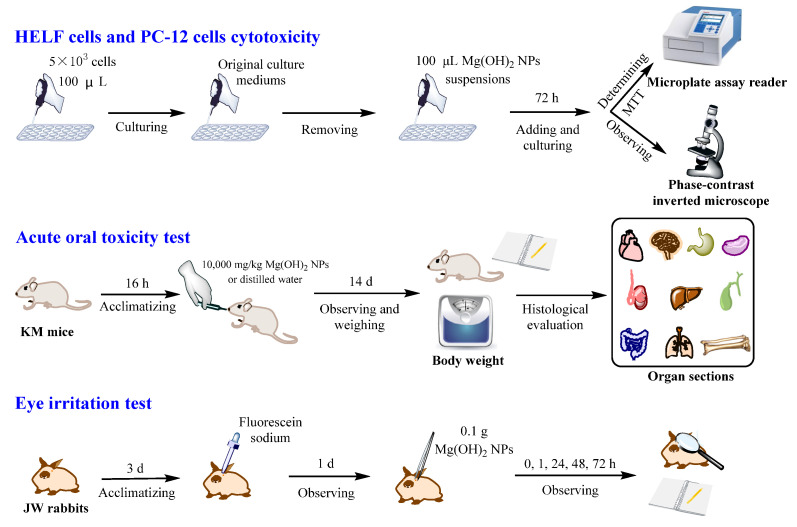
Experimental design.

**Figure 2 jfb-14-00229-f002:**
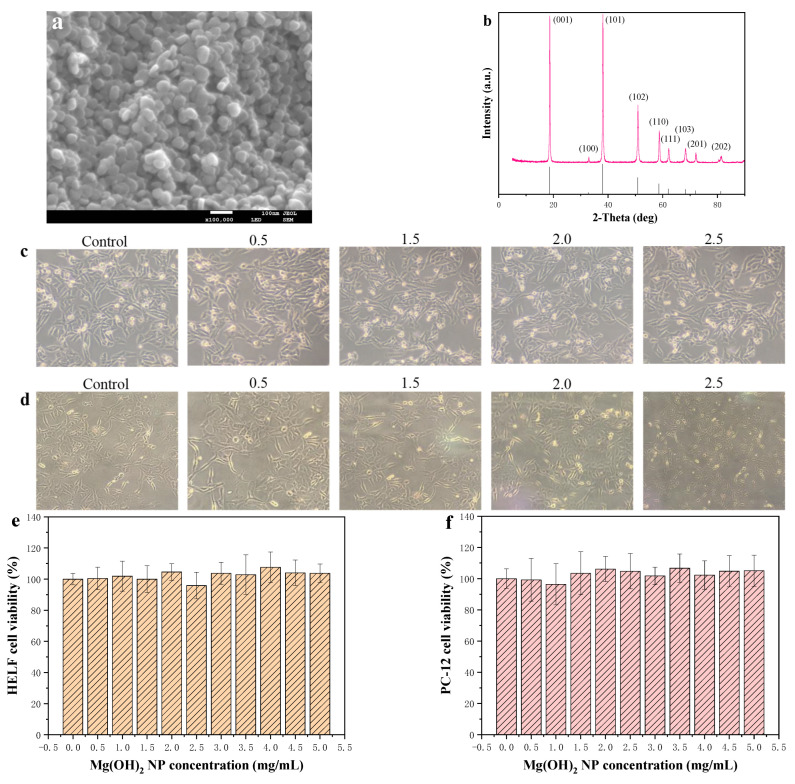
(**a**) SEM image of Mg(OH)_2_ NPs; (**b**) XRD pattern of Mg(OH)_2_ NPs; Cell morphology of (**c**) HELF and (**d**) CP-12; Cell viability of (**e**) HELF and (**f**) CP-12; (**g**) Histopathological sections.

**Figure 3 jfb-14-00229-f003:**
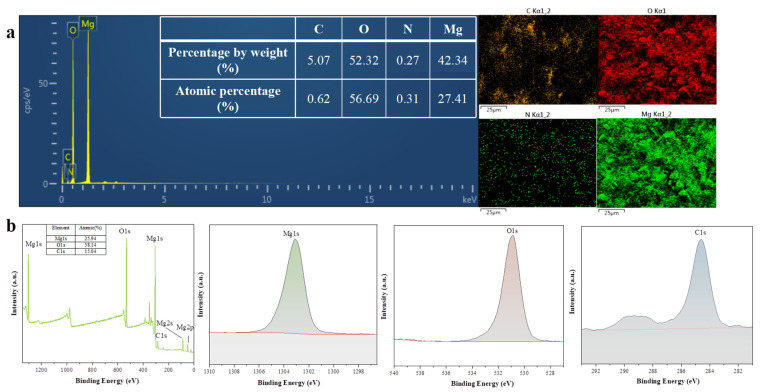
(**a**) EDS of Mg(OH)_2_ NPs (**b**) XPS of Mg(OH)_2_ NPs.

**Figure 4 jfb-14-00229-f004:**
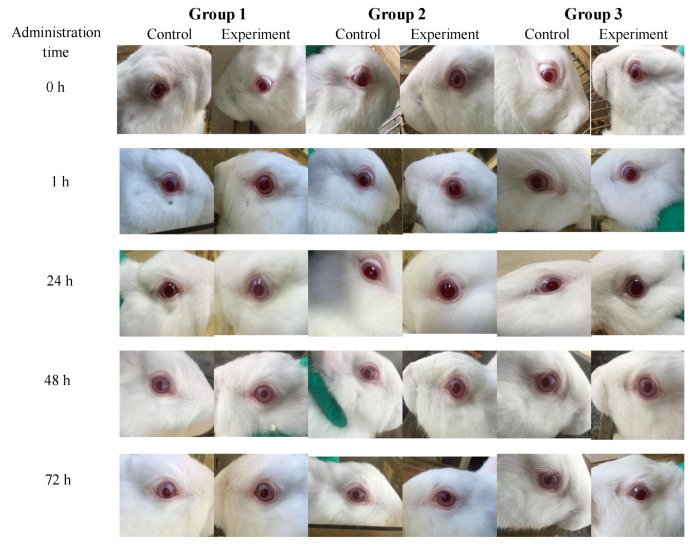
The eye images of JW rabbits.

**Table 1 jfb-14-00229-t001:** Score criteria of acute eye irritation.

Test Site		Appearance	Score
Cornea	A: Opacity	Same as normal eye	0
		Scattered or diffused turbidity; iris is clearly visible	1
		Translucent area is easy to distinguish; iris is clearly visible	2
		Appearance of a milky area; iris detail is unclear; pupil is barely visible	3
		Cornea is opaque; iris is unrecognizable	4
	B: Damaged area	0	0
		0~1/4	1
		1/4~1/2	2
		1/2~3/4	3
		3/4~1	4
		Integration 1 = A × B × 5, the maximum value is 80	
Iris		Same as normal eye	0
		Pleat is deepened/hyperemia/edema; pupil can respond to light	1
		Hyperemia/visible necrosis/pupil; cannot respond to light	2
		Integration 2 = A × 5, the maximum value is 10	
Conjunctiva	A: Hyperemia	Same as normal eye	0
		Degree of hyperemia is higher than amount of normal blood vessels	1
		Diffused dark-red hyperemia; blood vessels are hard to distinguish	2
		Diffused fuchsia hyperemia	3
	B: Edema	Same as normal eye	0
		Edema is more severe than normal edema	1
		Obvious edema and partial valgus eyelid	2
		Nearly half of eyelid closed caused by edema	3
		More than half of eyelid closed caused by edema	4
	C: Eye secretion	Same as normal eye	0
		Eye secretions are higher those of normal eyes	1
		Eyelids and eyelashes becoming wet due to eye secretions	2
		Large area of eyelids and around eye becoming wet due to eye secretions	3
		Integration 3 = (A + B + C) × 2; the maximum value is 20	
Total integration = Integration 1 + Integration 2 + Integration 3			

**Table 2 jfb-14-00229-t002:** Score of acute eye irritation.

JW Rabbits Samples (Acute Eye Irritation)	0 h	1 h	24 h	48 h	72 h
1	0	0	0	0	0
2	0	0	0	0	0
3	0	0	0	0	0
4	0	0	0	0	0

Acute eye irritation evaluation: no irritant.

## Data Availability

Not applicable.

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
