# Peer review of "Investigation of the Biosafety of Antibacterial Mg(OH)2 Nanoparticles to a Normal Biological System"

_jfb, 2023, doi:10.3390/jfb14040229_

Round 1

Reviewer 1 Report

  The manuscript describes assessment of Mg(OH)2 nanoparticle (NP) biosafety by using in vivo models, such as mice and rabbits.  The results are clear, which is that there is no toxic effect of Mg(OH)2 NP under high-dose conditions.  The experiments both by in vivo and culture models can be good examples for evaluation of their biosafety.  However, accurate characterization of Mg(OH)2 NP and comparison of the results obtained to references should be needed.  The manuscript can be published in JFB after revision based on the comments listed below.

1.        The authors should add references on the preparations of Mg(OH)2 NP used.  I did not judge whether the preparation is general or specific or not.  The toxic effect of inorganic NPs depends on their size and shapes.  The authors should show this point in result section.  Furthermore, the particles obtained seems large from the scale bar in Fig.1(a). The flaw feels readers unattractive to the manscript.  As well as the modification of figure, a raw SEM image of Mg(OH)2 NP should be sent. 

2.        The results obtained should be quantitatively compared to references both on Mg(OH)2 biosafety in vivo and culture models.  The results obtained show no toxic effect of Mg(OH)2 NP, and, in other words, no change after Mg(OH)2 NP dose.  In addition to that, how about comparison to other metal oxide NPs used in biotechnology?   

3.        Parts of discussion seem tedious because they should be described in introduction section.  I agree the author’s opinion that in vivo model using rats and rabbits are more accurate assessment than that of cell culture models.  However, more discussion based on results and comparison is needed.  The authors should rearrange the content of the manuscript including Fig.4.

Author Response

For reviewer 1:

The manuscript describes assessment of Mg(OH)2 nanoparticle (NP) biosafety by using in vivo models, such as mice and rabbits. The results are clear, which is that there is no toxic effect of Mg(OH)2 NP under high-dose conditions.  The experiments both by in vivo and culture models can be good examples for evaluation of their biosafety.  However, accurate characterization of Mg(OH)2 NP and comparison of the results obtained to references should be needed. The manuscript can be published in JFB after revision based on the comments listed below.

1.The authors should add references on the preparations of Mg(OH)2 NP used. I did not judge whether the preparation is general or specific or not.  The toxic effect of inorganic NPs depends on their size and shapes. The authors should show this point in result section.  Furthermore, the particles obtained seems large from the scale bar in Fig.1(a). The flaw feels readers unattractive to the manscript. As well as the modification of figure, a raw SEM image of Mg(OH)2 NP should be sent. 

Thank you very much for your valuable and thoughtful comments. We have completed the article according to your advise, and the revised manuscript is shown as below:

(Line 81-91, page 2)

 2.2. Synthesis of Mg(OH)2 NPs

The Mg(OH)2 NPs were prepared from the co-precipitation of MgCl2·6H2O, PEG and NH3·H2O at 60 oC for 1.5 h[26-27]. The 200 mL MgCl2 solutions (1.5 mol/L) were added to a 500 mL three-mouth flask, 0.4 g PEG was added as dispersant, and the three-mouth flask was placed in an ultrasonic bath to accelerate the dissolution of PEG. Then 5 wt% NH3·H2O was added slowly dropwise into the mixture (n(OH-)/n(Mg2+)≥2). The mixture was stirred at 300 rpm at 40℃ for 1.5 h. The resulting suspension was cooled to room temperature (20 oC) and aged for 12 h to allow complete precipitation of Mg2+. The resulting suspension was filtered and washed with ionized water and anhydrous ethanol, respectively. It was dried under vacuum at 80 oC for 4 h and ground uniformly to obtain the product nano-Mg(OH)2[26-27].

(Line 98-103, page 2)

2.3. Characterization

The elemental composition of the sample was tested by the scanning electron microscope (EDS, X-Max50, Oxford Instruments, UK) and X-ray photoelectron spectroscopy (XPS, Thermo ESCALAB 250Xi, Al K-Alpha) on a VG MultiLab 2000 X-ray photoelectron spectrometer (Thermo Electron Corporation) using Al-Kα (hλ=1486.6 eV) radiation as the excitation source, and the spectra were calibrated by the C 1s peak (284.8 eV).

(Line 177-194, page 5-6)

3.1. Characterizations of Mg(OH)2 NPs.

The composition of the surface elements of the synthesized Mg(OH)2 NPs products was studied by EDS spectrometer, and the EDS representing the strong characteristic peaks of magnesium and oxygen elements were shown. No other impurity elements were observed in the spectrum, indicating the high purity of the nanoparticle structure. The surface scanning of elements C, O, N and Mg is shown in Figure. 2c. The elements are presented with different colors respectively. The atomic percentage ratio of Mg and O elements was 27.41:56.69, and the atomic number ratio of Mg and O elements was close to 1:2. The weight percentages of Mg and O in the product were 42.34 and 52.32, respectively, which were consistent with the theoretical values. The EDS spectra showed that the synthesized product was Mg(OH)2 NPs.

XPS was used to analyze the surface chemical composition of Mg(OH)2 NPs. Figure 2d is the full-scan spectrum of the sample, which shows that the test sample mainly contains the characteristic peaks of Mg, O and C. No miscellaneous peaks of other substances appeared in the Figure. 2d, indicating high purity of the product. The XPS atlas shows that the characteristic peak of Mg2p appears at 49.6 eV and Mg1s at 1303.14eV. In the sample, Mg2p fits only one peak, which should be attributed to Mg2+ in Mg(OH)2. The characteristic peak of O1s appears at 531.12eV, which can also indicate the high purity of the generated Mg(OH)2 NPs.

(Line 245-249, page 7)

Figure 2. a. SEM image; b.XRD pattern; c-d.EDS and XPS of Mg(OH)2 NPs.

2.The results obtained should be quantitatively compared to references both on Mg(OH)2 biosafety in vivo and culture models. The results obtained show no toxic effect of Mg(OH)2 NP, and, in other words, no change after Mg(OH)2 NP dose. In addition to that, how about comparison to other metal oxide NPs used in biotechnology?   

Thank you very much for your careful reading of our manuscript, the comparison to other metal oxide NPs such as ZnO NPs and CuO NPs used in biotechnology have been analyzed in the section of introduction “In addition, Mg(OH)2 NPs have less cytotoxic effect on human cardiac microvascular endothelial cells than that of ZnO NPs and CuO NPs”. However, we didn't find the references both on Mg(OH)2 biosafety in vivo and culture models. Your significant valuable comments benefit us a lot, and the Mg(OH)2 biosafety in vivo and culture models will be researched in our next work.

3.Parts of discussion seem tedious because they should be described in introduction section. I agree the author’s opinion that in vivo model using rats and rabbits are more accurate assessment than that of cell culture models.  However, more discussion based on results and comparison is needed. The authors should rearrange the content of the manuscript including Fig.4.

Thank you very much for your valuable and thoughtful comment. We have rearranged the content of the manuscript including Fig.4, the revised manuscript is shown as below:

(Line 207-212, page 6)

3.3. Acute oral toxicity and histopathology research

The acute oral toxicity test can find out the lethal dose of the tested chemical, and the preliminary evaluation of the toxic effect characteristics, target organs, dose-response relationship and the risk of damage to human body of nano-Mg(OH)2 by observing the toxic effect performance, intensity of toxic effect and death of animals, the method is reproducible and can be used for acute toxicity evaluation with fewer animals. 

(Line 220-244, page 6-7)

The most direct way to diagnose pathological changes in organs is to prepare pathological tissue sections. Histological sections of KM mice after 14 d of transoral administration were analyzed in Figure. 3e. Pathological sections of all organs in the experimental group showed no abnormal changes compared with the control group. In the heart tissue, there was no inflammatory infiltration between cardiac myocytes, no myocarditis, and no granulation, vacuolation or fatty degeneration. Brain tissue sections showed no demyelination and inflammatory infiltrates in the brain and no abnormalities in astrocytes. There was no inflammatory cell infiltration in the lungs, no fibroplasia around the small airways, and no fibrosis in the interstitial lung. There were no signs of fibrosis in the myocardial fibers, much less degeneration, fragmentation, or vitreous changes. There was no blockage of blood vessels, no capillary abnormalities, and no swelling and detachment of endothelial cells. No inflammatory infiltrates, no interstitial inflammation, and no glomerular lesions, glomerular vascular tract cells and endothelial cells proliferation in the kidney tissues. Stomach tissue was intact with normal morphology of mural and principal cells. No necrosis was observed in the liver tissue and no high concentration of leukocytes was seen in the connective tissue, which indicates that no inflammation was generated in the liver after transoral administration. The size of lymphoid follicles in the spleen tissue of the experimental group was normal and comparable to that of the control group. Pathological sections of the large intestinal organs showed normal intestinal structure without infiltration of lymphocytes and plasma cells in the lamina propria, and the villi on the surface of the large intestine were not swollen and congested, and the intestinal glands did not bleed. In conclusion, the brain, heart, lung, kidney, stomach, liver and spleen of Mg(OH)2 nps-treated KM mice behaved normally without signs of necrosis or inflammation. Therefore, Mg(OH)2 NPs were less toxic to the above vital organs after 14 d of administration.

Reviewer 2 Report

Comments on the article:

“Investigation on the biosafety of antibacterial Mg(OH)2 nanoparticles to a normal biological system”

General comments: 

·       English grammar and spelling have to be extensively reviewed; Major rewriting is needed;

·       Please carefully justify the choice of animals and cells used in this study as well as the used tests: The Draize test method is recommended only as last resort -do you have any data concerning prior tests made with cells? There is no justification in this paper for the use of this last resort test and it should be provided. I would recommend to further elaborate this aspect in introduction with the works referred in order to justify the use of animals tests; i.e., the lack of toxicity in vitro tests performed by previous authors previously;

·        I would recommend to alter the order of in the protocol for the materials and methods; results and discussion: address the work on cells first and the animal tests later;

·       Please check all units of mass used – a uniform representation of concentration unit should be ideally used “e.g. 10 000 mg/kg” was administered to mousses but the masses for the in vitro cytotoxicity tests were expressed in “mg/ml”; Change where needed. 

·       The results do not support that 30-50 nm particles were used in this study; more tests/results are needed;

·       Please complete all captions in all figures used;

·       Figure 4 is not mentioned in the text; Also it is not clear what does it add to the overall content of the article;

Please take a look into these articles in order to rewrite major sections of materials and methods, results and discussion:

  • 10.3390/ijms23052612
  • 10.1016/j.tiv.2019.05.015
  • 10.3109/17435390.2015.1113321

Specific corrections:

Line 26: If NPs refers to Mg(OH)2 nanoparticles, as indicated in the abstract, then the bibliography listed does not concern these particles – please review or rewrite. Also add more information regarding these previous works: what were the major findings and current challenges?

Line 29, 30: please check the font size;

Line 56: please refer the method of preparation of NPs;

Line 57,58,59: please add more details regarding the acute oral toxicity and eye irritation test, namely, in which animals they were performed;

Line 62,63: please rephrase and refer to what has been generally shown by your work;

Line 66: please describe in more detail the method of co-precipitation – what quantities were used of each reagent; what were the steps, was there stirring? – if too extensive, I recommend a new section to be created for this, right after “2.1 – Materials”;

Line 78: Change to “Characterization”;

Line 80: please add the units for λ

Line 81 to 84: Please rewrite or rephrase in a clearer manner… was gold coating done for both XRD and SEM analysis or only SEM analysis? 

Line 88: please refer your solvent;

Line 89: what criteria were used to evaluate signs of clinical toxicity and abnormal behavior – please refer to the standards and evaluation criteria used;

Line 89-90: Please rephrase to justify the references placed in these sentences: why are these two references here? Was the method of evaluation of nanoparticle toxicity based on these two works? If so, please state it clearly;

Line 94: please state more clearly the concentration range of ethanol solutions and also refer the slicing procedure [BG1] used to obtain the longitudinal sections of the excised organs;

Line 98: Why “also”? Did you use any other method besides the “Draize method” to assess eye irritation? If so, refer to it. If not please remove “also”. Also check General comments about this method;

Line 99: please correct to “eye” instead of “eyes”; Please state clearly what solution was used for the drops (concentration and solvent); How many drops were added? If it was a solid powder (as the value indicated is expressed in grams), please remove the word “drops” and state so.

Line 102: animals have to be assessed at least twice daily, for the first three days after administration of the NPs; please state so;

Line 103: There is no information about the study of animal discomfort signs (repeated pawing of rubbing of the eye, excessive blinking or excessive tearing) in the protocol. Please add;

Line 105: Table 1 is not sufficiently clear; try to make it clear, for example, by adding extra horizontal lines to divide distinct sections and subsections such as between A: Opacity, B: Damaged area and integration value calculation in Cornea;

Line 109: was any antibiotic added to the medium? If not, why? If so, please state it. Why is reference 29 placed here? If some part of the method used is based in that work, please state so, but only if it is unique. What was the procedure to maintain the stock culture of the cells?

Line 111-112: Please describe in more detail the stock culture maintenance (number of passages and when);

Line 115: why the reference 30-31? When doing a colorimetric test for cell viability the determination is assumed to be calculated by this, this is part of the test – no need for this reference (the Eq1, line 118, is also not really needed but it can remain); 

Figure 1: The term “in vivo” should be used for both acute oral toxicity test and skin and irritation test, or not used in either, since they both use animals; If the term “in vivo” applied in animal tests, then the term “in vitro” should be applied to the cells tests; Please create a more complete caption for the figure describing what we are seeing. For instance, add a) acute oral toxicity tests; and so on describing in a succinct way, each test;

Line 125: The SEM micrograph of Fig. 2a) does not show 30-50 nm particles; we see large clusters with a size superior of several microns (large micro particles); Please place your measurements – this image does not show that the particles used; 

Line 127: Define JCPDS and further comment on the chemical identification of the samples;

Line 135: the method for evaluation of skin, hair and eye secretions is not described in materials and methods;

Line 140: Please rephrase; Based on the results, toxic dose of Mg(OH)2 is higher than the one used in this study, if applied orally, what do you mean by the appropriate dose?

Line 142-149: Description of histological sections is poor;

Figure 2: Similarly, to Figure 1, please complete the captions; The SEM micrograph does not show 30-50 nm particles; Accordingly identify the peaks of your XRD diagram; all microscope pictures are two small; The cells pictures taken from the phase contrast inverted microscope are missing the measurement bar;

Line 156: There is no score present in Figure 3 about for the eye irritation test; 

Lines 156-162: please add more comments to your results;

Lines 165-170: please further elaborate; Fig 2.e does not refer to morphology of HELF cells; 

Lines 179-184: repetition from what is stated in introduction; focus on the antibacterial mechanism of action and why they are concern for biosafety reasons; 

Lines 193-194: repetition from introduction, as stated previously the results shown do not demonstrate that 30-50 nm particles were used; The aim should be in the final paragraph of introduction and not in this section;

Lines 193-197: yes, but embryonic lung cells and neuronal cells are used in other tests reported here, which address this issue; It should be justified why choose the preformed tests in one paragraph; On the other hand, the acute oral toxicity test has its advantages over in vitro cell tests; state why the acute oral toxicity test?

Lines 208-210: repetition from introduction and Lines 174-184; Justify everything in one paragraph;

Lines 210-214: the data presented in this paper does not show nothing like that; It is still needed a justification for the performance of the Dazier test;

Lines 217-218: please rewrite; Place in introduction;

Lines 219-224: what is the relationship between lung interstitial fibrosis and the nanoparticles used in this study?

Line 236-240: please rewrite; it is not clear;

Lines 244-249: please rewrite, it is not clear;

Figure 4: is not mentioned in the text; does not seem to add anything to the content of the article;

Author Response

For reviewer 2:

1.English grammar and spelling have to be extensively reviewed; Major rewriting is needed.

Thank you very much for your careful reading of our manuscript. We have carefully proofread the manuscript and invited a native English speaker to help polish the article. The revised portions were marked in red in the revised manuscript.

2.Please carefully justify the choice of animals and cells used in this study as well as the used tests: The Draize test method is recommended only as last resort -do you have any data concerning prior tests made with cells? There is no justification in this paper for the use of this last resort test and it should be provided. I would recommend to further elaborate this aspect in introduction with the works referred in order to justify the use of animals tests; i.e., the lack of toxicity in vitro tests performed by previous authors previously.

Thank you very much for your significant comments and suggestions. However, we have no data concerning prior tests made with cells previously. As you said, it would be better if we provide the toxicity in vitro tests. Your significant valuable comments benefit us a lot, and the data concerning prior tests made with cells will be added in our next research.

3.I would recommend to alter the order of in the protocol for the materials and methods; results and discussion: address the work on cells first and the animal tests later.

Thank you very much for your valuable and thoughtful suggestion. The “order of in the protocol for the materials and methods; results and discussion” was adjusted as you said, and the revised manuscript is shown as below:

(Line 81, 92, page 2Line 104, 123, page 3;Line 143, page 4)

2.2. Synthesis of Mg(OH)2 NPs

2.3. Characterization

2.4. Culture and treatment of HELF cells and PC-12 cells

2.5. Acute oral toxicity test and histopathological sections

2.6. In vivo acute eye irritation test

4.Please check all units of mass used – a uniform representation of concentration unit should be ideally used “e.g. 10 000 mg/kg” was administered to mousses but the masses for the in vitro cytotoxicity tests were expressed in “mg/ml”; Change where needed. 

Thank you very much for your careful reading of our manuscript, and we have checked all units of mass used.

5.The results do not support that 30-50 nm particles were used in this study; more tests/results are needed.

Thanks for your careful reading of our manuscript, we were so sorry for this mistake. We have completed a new SEM characterization of Mg(OH)2 NPs used in this research. The revised manuscript is shown as below:

(Line 245, page 7)

6.Please complete all captions in all figures used.

Thanks again for your careful reading of our research, we have completed all captions in all figures used. The revised portions were marked in red in the revised manuscript.

7.Figure 4 is not mentioned in the text; Also it is not clear what does it add to the overall content of the article.

Thank you very much for the warning, we have deleted the Figure 4 in our manuscript.

8.Please take a look into these articles in order to rewrite major sections of materials and methods, results and discussion: 10.3390/ijms23052612; 10.1016/j.tiv.2019.05.015; 10.3109/17435390.2015.1113321. 

Thank you very much for your articles. We have read these articles carefully and revised our manuscript with reference to these articles.

9.Line 26: If NPs refers to Mg(OH)2 nanoparticles, as indicated in the abstract, then the bibliography listed does not concern these particles – please review or rewrite. Also add more information regarding these previous works: what were the major findings and current challenges?

Thank you very much for your valuable evaluations and suggestion. We have rewrote the References and added the major findings and current challenges in our manuscript according to your advise.

10.Line 29, 30: please check the font size.

Thanks for your careful reading of our manuscript. We were so sorry for this mistake, and we have already seriously read the manuscript and corrected the error.

11.Line 56: please refer the method of preparation of NPs.

Thank you very much for your valuable evaluation, we have cited References used for the method of preparation of NPs. The revised manuscript is shown as below:

[26] Zhang, W.T., Zhang, P.C., Wang, Y.L., Li, J.F., 2015. Preparation of Mg(OH)2 Nanosheets and Self-Assembly of Its Flower-Like Nanostructure via Precipitation Method for Heat-Resistance Application. Integrated Ferroelectrics. 163, 148-154.

[27] Wang, P.P., Li, C.H., Gong, H.Y., Wang, H.Q., Liu, J.R., 2011. Morphology control and growth mechanism of magnesium hydroxide nanoparticles via a simple wet precipitation method. Ceram. Int. 37, 3365-3370.

12.Line 57,58,59: please add more details regarding the acute oral toxicity and eye irritation test, namely, in which animals they were performed.

Thank you very much for your valuable advice, the more details regarding the acute oral toxicity and eye irritation test were add in our manuscript as you said. The revised manuscript is shown as below:

(Line 123, page 3)

2.5. Acute oral toxicity test and histopathological sections

The acute oral toxicity experiment was performed in a constant temperature animal laboratory (25 ℃), and the KM mice were kept for 3 d to adapt to the laboratory environment, while the growth and disease status of KM mice were observed (GBZ/T 240.2-2011). As shown in Figure. 1, 20 healthy KM mice without abnormal behavioral signs were divided into two groups and fasted for 16 h. The control group was given sterile water (The solvent was sterile deionized water), and the experimental group was given a suspension of 10000 mg/kg of Mg(OH)2 NPs in sterile water as a solvent for one-time oral gavage of KM mice. After administration, the mice were normally housed and observed for abnormalities in skin condition, hair volume, respiratory rate and eye secretions at 0, 4, 7, 11 and 14 d, especially for clinical signs of toxicity such as coma and drowsiness. Mortality was recorded after 14 d of treatment according to some researches[26-27].

(Line 143, page 4)

2.6. In vivo acute eye irritation test

The Draize method was used for acute eye irritation toxicity experiment. The animals have to be assessed at least twice daily, for the first three days after administration of the Mg(OH)2 NPs. The JW rabbits were kept in a constant temperature animal laboratory for 3 d before the experiment to adapt to the laboratory environment, while the growth and disease status were observed. As shown in Figure. 1, the eye of four healthy, disease-free JW rabbits were selected and kept for 3 d. Then, both eye of each JW rabbit were examined to confirm the absence of signs of eye irritation (conjunctival rupture, corneal defects and iris damage). The JW rabbits were fixed, the left lower eyelid was gently pulled down, and Mg(OH)2 NPs (0.1 g) were immediately added into the conjunctival sac and the eye were closed for 1 min. the untreated right eye was used as a control. After 0, 1, 24, 48, and 72 h of administration, the conjunctiva, cornea, and iris of the JW rabbit eye were examined with magnification and recorded for abnormalities. Table 1 shows the scoring criteria for acute ocular irritation. There was no animal discomfort sign including repeated pawing of rubbing of the eye, excessive blinking or excessive tearing during this process.

13.Line 62,63: please rephrase and refer to what has been generally shown by your work.

Thank you very much for your valuable suggestion. We have rephrased and referred to what has been generally shown by our work. The revised manuscript is shown as below:

(Line 63-65, page 2)

Additionally, the cytotoxicity effect of Mg(OH)2 NPs on respiratory and brain nervous systems was evaluated through the in vitro test of human embryonic lung fibroblasts (HELF) cells and rat adrenergic neural tumour phaeochromocytoma (PC-12) cells.

14.Line 66: please describe in more detail the method of co-precipitation – what quantities were used of each reagent; what were the steps, was there stirring? – if too extensive, I recommend a new section to be created for this, right after “2.1 – Materials”.

Thank you very much for your valuable and thoughtful comment, we have describe in more detail the method of co-precipitation in our work. The revised manuscript is shown as below:

(Line 81-90, page 2)

2.2. Synthesis of Mg(OH)2 NPs

The Mg(OH)2 NPs were prepared from the co-precipitation of MgCl2·6H2O, PEG and NH3·H2O at 60 oC for 1.5 h[26-27]. The 200 mL MgCl2 solutions (1.5 mol/L) were added to a 500 mL three-mouth flask, 0.4 g PEG was added as dispersant, and the three-mouth flask was placed in an ultrasonic bath to accelerate the dissolution of PEG. Then 5 wt% NH3·H2O was added slowly dropwise into the mixture (n(OH-)/n(Mg2+)≥2). The mixture was stirred at 300 rpm at 40℃ for 1.5 h. The resulting suspension was cooled to room temperature (20 oC) and aged for 12 h to allow complete precipitation of Mg2+. The resulting suspension was filtered and washed with ionized water and anhydrous ethanol, respectively. It was dried under vacuum at 80 oC for 4 h and ground uniformly to obtain the product nano-Mg(OH)2[26-27].

15.Line 78: Change to “Characterization”.

Thank you very much for your valuable suggestion, the “Characterizations was changed to “Characterization”.

16.Line 80: please add the units for λ. 

Thank you very much for your warning, we were so sorry for this mistake. We have already added the units for λ (nm). The revised manuscript is shown as below:

17.Line 81 to 84: Please rewrite or rephrase in a clearer manner… was gold coating done for both XRD and SEM analysis or only SEM analysis?

Thanks again for your warning, we were so sorry for this error. We have already rephrased in a clearer manner. The revised manuscript is shown as below:

(Line 98-99, page 3)

In addition, the conductive films with gold were coated on the Mg(OH)2 NPs surface for SEM test.

18.Line 88: please refer your solvent.

Thanks for your thoughtful comment. We have referred our solvent in this research, and the revised manuscript is shown as below:

(Line 129, page 3)

The solvent was sterile deionized water.

19.Line 89: what criteria were used to evaluate signs of clinical toxicity and abnormal behavior – please refer to the standards and evaluation criteria used.

Thank you very much for your significant suggestion, we have added the standard in our manuscript.

(Line 207-212, page 6)

3.3. Acute oral toxicity and histopathology research

The acute oral toxicity test can find out the lethal dose of the tested chemical, and the preliminary evaluation of the toxic effect characteristics, target organs, dose-response relationship and the risk of damage to human body of nano-Mg(OH)2 by observing the toxic effect performance, intensity of toxic effect and death of animals, the method is reproducible and can be used for acute toxicity evaluation with fewer animals. 

(Line 220-244, page 6-7)

The most direct way to diagnose pathological changes in organs is to prepare pathological tissue sections. Histological sections of KM mice after 14 d of transoral administration were analyzed in Figure. 3e. Pathological sections of all organs in the experimental group showed no abnormal changes compared with the control group. In the heart tissue, there was no inflammatory infiltration between cardiac myocytes, no myocarditis, and no granulation, vacuolation or fatty degeneration. Brain tissue sections showed no demyelination and inflammatory infiltrates in the brain and no abnormalities in astrocytes. There was no inflammatory cell infiltration in the lungs, no fibroplasia around the small airways, and no fibrosis in the interstitial lung. There were no signs of fibrosis in the myocardial fibers, much less degeneration, fragmentation, or vitreous changes. There was no blockage of blood vessels, no capillary abnormalities, and no swelling and detachment of endothelial cells. No inflammatory infiltrates, no interstitial inflammation, and no glomerular lesions, glomerular vascular tract cells and endothelial cells proliferation in the kidney tissues. Stomach tissue was intact with normal morphology of mural and principal cells. No necrosis was observed in the liver tissue and no high concentration of leukocytes was seen in the connective tissue, which indicates that no inflammation was generated in the liver after transoral administration. The size of lymphoid follicles in the spleen tissue of the experimental group was normal and comparable to that of the control group. Pathological sections of the large intestinal organs showed normal intestinal structure without infiltration of lymphocytes and plasma cells in the lamina propria, and the villi on the surface of the large intestine were not swollen and congested, and the intestinal glands did not bleed. In conclusion, the brain, heart, lung, kidney, stomach, liver and spleen of Mg(OH)2 nps-treated KM mice behaved normally without signs of necrosis or inflammation. Therefore, Mg(OH)2 NPs were less toxic to the above vital organs after 14 d of administration.

  1. Line 89-90: Please rephrase to justify the references placed in these sentences: why are these two references here? Was the method of evaluation of nanoparticle toxicity based on these two works? If so, please state it clearly.

(Line 123-135, page 3)

2.5. Acute oral toxicity test and histopathological sections

The acute oral toxicity experiment was performed in a constant temperature animal laboratory (25 ℃), and the KM mice were kept for 3 d to adapt to the laboratory environment, while the growth and disease status of KM mice were observed (GBZ/T 240.2-2011). As shown in Figure. 1, 20 healthy KM mice without abnormal behavioral signs were divided into two groups and fasted for 16 h. The control group was given sterile water (The solvent was sterile deionized water), and the experimental group was given a suspension of 10000 mg/kg of Mg(OH)2 NPs in sterile water as a solvent for one-time oral gavage of KM mice. After administration, the mice were normally housed and observed for abnormalities in skin condition, hair volume, respiratory rate and eye secretions at 0, 4, 7, 11 and 14 d, especially for clinical signs of toxicity such as coma and drowsiness. Mortality was recorded after 14 d of treatment according to some researches[26-27].

21.Line 94: please state more clearly the concentration range of ethanol solutions and also refer the slicing procedure [BG1] used to obtain the longitudinal sections of the excised organs.

Thank you very much for your significant comments and suggestions. The operation process has been described in a new table, the revised manuscript is shown as below:

(Line 136-142, page 3)

After the acute oral toxicity test, KM mice were executed by cervical dislocation method and analyzed for histopathological changes after drug administration. Excised organs were fixed in 4% paraformaldehyde, decalcified in EDTA, and then dehydrated in a series of graded ethanol (70%、80%、95% and 100%) and embedded in paraffin to prepare tissue sections. Longitudinal sections of organs (5 µm thick) were stained with hematoxylin-eosin (HE) and Giemsa staining, and pathological changes in organ tissue were observed with transmission light microscopy (CKX41)[28].

22.Line 98: Why “also”? Did you use any other method besides the “Draize method” to assess eye irritation? If so, refer to it. If not please remove “also”. Also check General comments about this method.

Thanks for your careful reading of our manuscript. We were so sorry for this mistake, and we have already seriously read the manuscript and corrected the error of “also”.

23.Line 99: please correct to “eye” instead of “eyes”; Please state clearly what solution was used for the drops (concentration and solvent); How many drops were added? If it was a solid powder (as the value indicated is expressed in grams), please remove the word “drops” and state so.

Thanks for your reminding. We have seriously corrected to “eye” instead of “eyes”. We have revised the section of “In vivo acute eye irritation test”, the revised manuscript is shown as below:

(Line 146-157, page 4)

The JW rabbits were kept in a constant temperature animal laboratory for 3 d before the experiment to adapt to the laboratory environment, while the growth and disease status were observed. As shown in Figure. 1, the eye of four healthy, disease-free JW rabbits were selected and kept for 3 d. Then, both eye of each JW rabbit were examined to confirm the absence of signs of eye irritation (conjunctival rupture, corneal defects and iris damage). The JW rabbits were fixed, the left lower eyelid was gently pulled down, and Mg(OH)2 NPs (0.1 g) were immediately added into the conjunctival sac and the eye were closed for 1 min. the untreated right eye was used as a control. After 0, 1, 24, 48, and 72 h of administration, the conjunctiva, cornea, and iris of the JW rabbit eye were examined with magnification and recorded for abnormalities. Table 1 shows the scoring criteria for acute ocular irritation.

24.Line 102: animals have to be assessed at least twice daily, for the first three days after administration of the NPs; please state so.

Thank you very much for your valuable and thoughtful comment, , the revised manuscript is shown as below:

(Line 144-146, page 4)

2.6. In vivo acute eye irritation test

The Draize method was used for acute eye irritation toxicity experiment. The animals have to be assessed at least twice daily, for the first three days after administration of the Mg(OH)2 NPs.

25.Line 103: There is no information about the study of animal discomfort signs (repeated pawing of rubbing of the eye, excessive blinking or excessive tearing) in the protocol. Please add.

Thank you very much for your valuable evaluation and suggestion. We have added the information about the study of animal discomfort signs (repeated pawing of rubbing of the eye, excessive blinking or excessive tearing) in the protocol. The revised manuscript is shown as below:

(Line 157-159, page 4)

There was no animal discomfort sign including repeated pawing of rubbing of the eye, excessive blinking or excessive tearing during this process.

26.Line 105: Table 1 is not sufficiently clear; try to make it clear, for example, by adding extra horizontal lines to divide distinct sections and subsections such as between A: Opacity, B: Damaged area and integration value calculation in Cornea.

Thank you very much for your reminding. We have revised the Table 1, and the revised manuscript is shown as below:

(Line 160, page 4)

Table 1. Score criteria of acute eye irritation.

Test site

Appearance

Score

Cornea

A: Opacity

Same as normal eye

0

Scattered or diffuse turbidity, iris is clearly visible

1

Translucent area is easy to distinguish, iris is clearly visible

2

Milky area is appear, iris detail is unclear, pupil is barely visible

3

Cornea is opaque, iris is unrecognizable

4

B: Damaged

area

0

0

0~1/4

1

1/4~1/2

2

1/2~3/4

3

3/4~1

4

Integration 1 = A×B×5, the maximum value is 80

Iris

Same as normal eye

0

Pleat is deepened/hyperemia/edema, pupil can respond to light

1

Hyperemia / visible necrosis / pupil can not respond to light

2

Integration 2 = A×5, the maximum value is 10

Conjunctiva

A: Hyperemia

Same as normal eye

0

Hyperemia amount is higher than that of normal blood vessels

1

Diffuse dark red hyperemia, blood vessels are hard to distinguish

2

Diffuse fuchsia hyperemia

3

B: Edema

Same as normal eye

0

Edema is significant than that of normal edema

1

Obvious edema and partial valgus eyelid

2

Nearly half of eyelid closed caused by edema

3

More than half of eyelid closed caused by edema

4

C: Eye

secretion

Same as normal eye

0

Eye secretions are higher than that of normal eye

1

Eyelids and eyelashes just getting wet cause by eye secretions

2

Large area of eyelids and around eye getting wet cause by eye secretions

3

Integration 3 = (A+B+C)×2, the maximum value is 20

Total integration = Integration 1+Integration 2+Integration 3

27.Line 109: was any antibiotic added to the medium? If not, why? If so, please state it. Why is reference 29 placed here? If some part of the method used is based in that work, please state so, but only if it is unique. What was the procedure to maintain the stock culture of the cells?

Thank you very much for your valuable suggestion. In this experiment, no antibiotics were added to the culture medium and cell culture methods were borrowed. We have revised the process in the revised manuscript:

(Line 104-107, page 3)

2.4. Culture and treatment of HELF cells and PC-12 cells

The HELF cells and PC-12 cells were maintained in Dulbecco's Modified Eagle Medium (DMEM) containing 10% fetal bovine serum as single cell suspensions after centrifugation at 1000 rpm and cultured at 37 oC[29] in a 5% CO2 humidified incubator.

28.Line 111-112: Please describe in more detail the stock culture maintenance (number of passages and when).

Thanks again for your reminding. We have revised the process, and the revised manuscript is shown as below:

(Line 107-120, page 3)

Mg(OH)2 NPs were added to DMEM to obtain a range of concentrations (0, 0.5, 1.0, 1.5, 2.0, 2.5, 3.0, 3.5, 4.0, 4.5, and 5.0 mg/mL) as shown in Figure. 1. In addition, 5×103 cells (100 µL, 3 generations, 24 h) per well were inoculated into 96-well plates. After cells were fully attached, the original medium were replaced with 100 µL Mg(OH)2 NPs suspensions or 100 µL normal complete culture media for the control group. After 72 h of incubation in the incubator, cell viability was determined by the 3-(4,5-dimethylthiazole)-2,5 diphenyltetrazolium bromide (MTT) method by adding 15 µL of 5 mg/mL MTT working solution per well and continuing incubation in the incubator for 4 h. Subsequently, 150 μL DMSO was added to each well, shaken for 10 min, and the cell viability was calculated by Eq. (1), and the optical densitometry (OD) values at 570 nm were measured using an enzyme plate tester (SpectraMax i3x). In addition, the morphology of the cells was observed under an inverted microscope (KOSTER IMC 800Ti).

29.Line 115: why the reference 30-31? When doing a colorimetric test for cell viability the determination is assumed to be calculated by this, this is part of the testno need for this reference (the Eq1, line 118, is also not really needed but it can remain). 

Thanks for your advice, we have carefully deleted the reference 30-31.

30.Figure 1: The term “in vivo” should be used for both acute oral toxicity test and skin and irritation test, or not used in either, since they both use animals; If the term “in vivo” applied in animal tests, then the term “in vitro” should be applied to the cells tests; Please create a more complete caption for the figure describing what we are seeing. For instance, add a) acute oral toxicity tests; and so on describing in a succinct way, each test.

Thanks again for your suggestion,we have created a more complete caption for the figure describing what you are seeing. The revised manuscript is shown as below:

(Line 162, page 5)

Figure 1. Experimental design.

31.Line 125: The SEM micrograph of Fig. 2a) does not show 30-50 nm particles; we see large clusters with a size superior of several microns (large micro particles); Please place your measurements – this image does not show that the particles used. 

Thank you very much for your valuable and thoughtful comments. We have completed the article according to your advise, and the revised manuscript is shown as below:

(Line 245, page 7)

(Line 177-194, page 5-6)

3.1. Characterizations of Mg(OH)2 NPs.

The composition of the surface elements of the synthesized Mg(OH)2 NPs products was studied by EDS spectrometer, and the EDS representing the strong characteristic peaks of magnesium and oxygen elements were shown. No other impurity elements were observed in the spectrum, indicating the high purity of the nanoparticle structure. The surface scanning of elements C, O, N and Mg is shown in Figure. 2c. The elements are presented with different colors respectively. The atomic percentage ratio of Mg and O elements was 27.41:56.69, and the atomic number ratio of Mg and O elements was close to 1:2. The weight percentages of Mg and O in the product were 42.34 and 52.32, respectively, which were consistent with the theoretical values. The EDS spectra showed that the synthesized product was Mg(OH)2 NPs.

XPS was used to analyze the surface chemical composition of Mg(OH)2 NPs. Figure 2d is the full-scan spectrum of the sample, which shows that the test sample mainly contains the characteristic peaks of Mg, O and C. No miscellaneous peaks of other substances appeared in the Figure. 2d, indicating high purity of the product. The XPS atlas shows that the characteristic peak of Mg2p appears at 49.6 eV and Mg1s at 1303.14eV. In the sample, Mg2p fits only one peak, which should be attributed to Mg2+ in Mg(OH)2. The characteristic peak of O1s appears at 531.12eV, which can also indicate the high purity of the generated Mg(OH)2 NPs.

32.Line 127: Define JCPDS and further comment on the chemical identification of the samples.

Thank you very much for the warning, the full name of the JCPDS was added in our manuscript. The revised manuscript is shown as below:

(Line 169, page 5)

Joint Committee on Powder Diffraction Standards (JCPDS 07-0239)

33.Line 135: the method for evaluation of skin, hair and eye secretions is not described in materials and methods.

Thank your very much for your thoughtful advise. The revised manuscript is shown as below: 

(Line 123-135, page 3)

2.5. Acute oral toxicity test and histopathological sections

The acute oral toxicity experiment was performed in a constant temperature animal laboratory (25 ℃), and the KM mice were kept for 3 d to adapt to the laboratory environment, while the growth and disease status of KM mice were observed (GBZ/T 240.2-2011). As shown in Figure. 1, 20 healthy KM mice without abnormal behavioral signs were divided into two groups and fasted for 16 h. The control group was given sterile water (The solvent was sterile deionized water), and the experimental group was given a suspension of 10000 mg/kg of Mg(OH)2 NPs in sterile water as a solvent for one-time oral gavage of KM mice. After administration, the mice were normally housed and observed for abnormalities in skin condition, hair volume, respiratory rate and eye secretions at 0, 4, 7, 11 and 14 d, especially for clinical signs of toxicity such as coma and drowsiness. Mortality was recorded after 14 d of treatment according to some researches[26-27].

34.Line 140: Please rephrase; Based on the results, toxic dose of Mg(OH)2 is higher than the one used in this study, if applied orally, what do you mean by the appropriate dose?

Thanks for your thoughtful comment, the “appropriate dose” was changed to “lethal dose”.

35.Line 142-149: Description of histological sections is poor.

Thank you very much for your valuable evaluations and suggestion. We have enrich the Description of histological sections, the revised manuscript is shown as below.

(Line 207-244, page 6-7)

3.3. Acute oral toxicity and histopathology research

The acute oral toxicity test can find out the lethal dose of the tested chemical, and the preliminary evaluation of the toxic effect characteristics, target organs, dose-response relationship and the risk of damage to human body of nano-Mg(OH)2 by observing the toxic effect performance, intensity of toxic effect and death of animals, the method is reproducible and can be used for acute toxicity evaluation with fewer animals. The skin, hair, respiration and eye secretions content of KM mice administered with Mg(OH)2 NPs were the same as those of the control group. And in 14 d, there was no clinical toxicity such as coma, diarrhea, lethargy, nausea, salivation, vomiting, convulsions and tremors, and the mortality ratio was 0%. Thus, the KM mice appeared to be normal with no sign of toxicity at a dose of 10000 mg/kg administered orally, and the lethal dose of Mg(OH)2 NPs could be more than 10000 mg/kg, which fell into the actual non-toxic range.

The most direct way to diagnose pathological changes in organs is to prepare pathological tissue sections. Histological sections of KM mice after 14 d of transoral administration were analyzed in Figure. 3e. Pathological sections of all organs in the experimental group showed no abnormal changes compared with the control group. In the heart tissue, there was no inflammatory infiltration between cardiac myocytes, no myocarditis, and no granulation, vacuolation or fatty degeneration. Brain tissue sections showed no demyelination and inflammatory infiltrates in the brain and no abnormalities in astrocytes. There was no inflammatory cell infiltration in the lungs, no fibroplasia around the small airways, and no fibrosis in the interstitial lung. There were no signs of fibrosis in the myocardial fibers, much less degeneration, fragmentation, or vitreous changes. There was no blockage of blood vessels, no capillary abnormalities, and no swelling and detachment of endothelial cells. No inflammatory infiltrates, no interstitial inflammation, and no glomerular lesions, glomerular vascular tract cells and endothelial cells proliferation in the kidney tissues. Stomach tissue was intact with normal morphology of mural and principal cells. No necrosis was observed in the liver tissue and no high concentration of leukocytes was seen in the connective tissue, which indicates that no inflammation was generated in the liver after transoral administration. The size of lymphoid follicles in the spleen tissue of the experimental group was normal and comparable to that of the control group. Pathological sections of the large intestinal organs showed normal intestinal structure without infiltration of lymphocytes and plasma cells in the lamina propria, and the villi on the surface of the large intestine were not swollen and congested, and the intestinal glands did not bleed. In conclusion, the brain, heart, lung, kidney, stomach, liver and spleen of Mg(OH)2 nps-treated KM mice behaved normally without signs of necrosis or inflammation. Therefore, Mg(OH)2 NPs were less toxic to the above vital organs after 14 d of administration.

36.Figure 2: Similarly, to Figure 1, please complete the captions; The SEM micrograph does not show 30-50 nm particles; Accordingly identify the peaks of your XRD diagram; all microscope pictures are two small; The cells pictures taken from the phase contrast inverted microscope are missing the measurement bar.

Thank you very much for your valuable and thoughtful comments. The revised manuscript is shown as below:

(Line 162, page 5)

Figure 1. Experimental design.

(Line 245, page 7)

37.Line 156: There is no score present in Figure 3 about for the eye irritation test. 

Thank you very much for your suggestion. The score of the eye irritation test is in Table 2.

38.Lines 156-162: please add more comments to your results.

Thanks again for your suggestion, we have added more comments to our results in the revised manuscript. Please review the revised manuscript.

(Line 163-272, page 5-9)

39.Lines 165-170: please further elaborate; Fig 2.e does not refer to morphology of HELF cells. HELF形态分析

Thank you very much for your valuable evaluations and suggestion. The revised manuscript is shown as below:

(Line 195-201, page 6)

3.2. In vitro cytotoxicity of Mg(OH)2 NPs to HELF cells and PC-12 cells 

As shown in Figure. 3a, the cell viability of HELF cells showed no obvious fluctuation after cells were treated with different concentrations (0, 0.5, 1.0, 1.5, 2.0, 2.5, 3.0, 3.5, 4.0, 4.5, 5.0 mg/mL) of Mg(OH)2 NPs, which indicateed that Mg(OH)2 NPs had no significant toxic effect on the proliferation of HELF cells. Additionally, the morphology of HELF cells in Figure. 3a remained normal after the treatment of different concentrations of Mg(OH)2 NPs.

40.Lines 179-184: repetition from what is stated in introduction; focus on the antibacterial mechanism of action and why they are concern for biosafety reasons. 

Thank you very much for your valuable suggestion, we have already deleted the section.

41.Lines 193-194: repetition from introduction, as stated previously the results shown do not demonstrate that 30-50 nm particles were used; The aim should be in the final paragraph of introduction and not in this section.

Thanks again for your valuable suggestion, we have already added a new SEM of Mg(OH)2 NPs and deleted the repetition.

(Line 58-67, page 2)

In this work, synthetic Mg(OH)2 NPs were characterized by scanning electron microscopy (SEM), X-ray diffraction (XRD), energy dispersive X-ray spectrometer (EDS) and X-ray photoelectron spectroscopy (XPS). The safe dose was evaluated by the acute oral toxicity test. The effect on vital organs was determined by the histopathology analysis. The irritation to eye was evaluated by the acute irritation test in vivo. Additionally, the cytotoxicity effect of Mg(OH)2 NPs on respiratory and brain nervous systems was evaluated through the in vitro test of human embryonic lung fibroblasts (HELF) cells and rat adrenergic neural tumour phaeochromocytoma (PC-12) cells. The aforementioned experiments comprehensively evaluated the biosafety of Mg(OH)2 NPs to a normal biological system.

42.Lines 193-197: yes, but embryonic lung cells and neuronal cells are used in other tests reported here, which address this issue; It should be justified why choose the preformed tests in one paragraph; On the other hand, the acute oral toxicity test has its advantages over in vitro cell tests; state why the acute oral toxicity test?

Thank you very much for your significant comments and suggestions. This part has been adjusted and the reason why the acute oral toxicity test was added in our article, the revised manuscript is shown as below:

(Line 208-212, page 6)

The acute oral toxicity test can find out the lethal dose of the tested chemical, and the preliminary evaluation of the toxic effect characteristics, target organs, dose-response relationship and the risk of damage to human body of nano-Mg(OH)2 by observing the toxic effect performance, intensity of toxic effect and death of animals, the method is reproducible and can be used for acute toxicity evaluation with fewer animals. 

43.Lines 208-210: repetition from introduction and Lines 174-184; Justify everything in one paragraph.

Thank you very much for your valuable suggestion, we have already deleted the section.

44.Lines 210-214: the data presented in this paper does not show nothing like that; It is still needed a justification for the performance of the Dazier test.

Thank you very much for your significant comments and suggestions. This part has been adjusted and the reason why the Dazier test was added in our article, the revised manuscript is shown as below:

(Line 258-262, page 8)

The acute eye irritation toxicity test (Draize test) was performed by contacting Mg(OH)2 NPs with the eye of JW rabbits, and the irritation level of magnesium hydroxide nanoparticles could be directly measured. Due to the sensitivity of the rabbit eye itself, the irritation effect of Mg(OH)2 NPs in human eye could be predicted, and the potential hazard of Mg(OH)2 NPs to human eye could be easily identified

45.Lines 217-218: please rewrite; Place in introduction.

Thank you very much for your valuable advice, we have rewrote the section of “introduction”.

46.Lines 219-224: what is the relationship between lung interstitial fibrosis and the nanoparticles used in this study?

Thank you very much for the valuable suggestion, we have explained the issue as below:

Nanoparticles can inhibit the phagocytic function of alveolar macrophages. When inhalation exceeds the lung load, the alveolar macrophage-mediated scavenging ability is weakened, and nanoparticles accumulate in the lung. Causes apoptosis of lung cells, inflammation, leading to pulmonary interstitial fibrosis[1-3].

[1]David, B.W., 2010. Assessing health risks of inhaled nanomaterials: development of pulmonary bioassay hazard studies, Anal. Bioanal. Chem. 398, 607–612.

[2]Miriam, T.K., Eva, M.G., Tobias, S., Carola, V., 2022. Lung Organoids for Hazard Assessment of Nanomaterials.Int. J. Mol. Sci.23,15666.

[3]Landsiedel, R., Ma-Hock, L., Treumann, S., Strauss, V., Wohlleben, W., Wiench, K., Ravenzwaay, B.V., 2010. Inhalation toxicity studies with 12 nanomaterials—Comparing effects and distribution in the lung. Toxicol. Lett. 196, S281.

47.Line 236-240: please rewrite; it is not clear.

Thank you very much for your valuable advice, we have rewrote the section.

48.Lines 244-249: please rewrite, it is not clear.

Thanks again for your valuable advice, we have rewrote the section.

49.Figure 4: is not mentioned in the text; does not seem to add anything to the content of the article.

Thank you very much for the warning, we have already deleted the Figure 4.

Reviewer 3 Report

This paper by Y. Wang and Y. Zhu reports the studies on the biosafety of Mg(OH)2 nanoparticles, with widely known antibacterial activity, to biological systems. The authors have performed in vivo tests to evaluate acute oral toxicity, which showed that Mg(OH)2 NPs did not induced toxicity neither affected vital organs of KM mice. Furthermore, in vitro tests showed that Mg(OH)2 NPs did not cause respiratory disease of pulmonary interstitial fibrosis and had no effect on dopaminergic neurons. The study was well conducted and the results appropriately analyzed, providing useful information regarding the future application of this type of material as antibacterial.

In my opinion, the article can be accepted for publication in Journal of Functional Biomaterials, after minor revision.

- More detailed procedure should be given to the synthesis/preparation of Mg(OH)2 nanoparticles. If the authors have used a literature procedure, the bibliographic references should be given in section 2.1.

- Regarding the NP’s characterization, additional EDS or XPS analysis could provide relevant information regarding the surface of the Mn(OH)2 NP’s, and their elemental composition.

- To my view, Table 2 is useless, since all runs and entries have the same (zero) value. A simple comment in the text seems sufficient.

Author Response

For reviewer 3:

This paper by Y. Wang and Y. Zhu reports the studies on the biosafety of Mg(OH)2 nanoparticles, with widely known antibacterial activity, to biological systems. The authors have performed in vivo tests to evaluate acute oral toxicity, which showed that Mg(OH)2 NPs did not induced toxicity neither affected vital organs of KM mice. Furthermore, in vitro tests showed that Mg(OH)2 NPs did not cause respiratory disease of pulmonary interstitial fibrosis and had no effect on dopaminergic neurons. The study was well conducted and the results appropriately analyzed, providing useful information regarding the future application of this type of material as antibacterial. In my opinion, the article can be accepted for publication in Journal of Functional Biomaterials, after minor revision.

1.More detailed procedure should be given to the synthesis/preparation of Mg(OH)2 nanoparticles. If the authors have used a literature procedure, the bibliographic references should be given in section 2.1.

Thank you very much for your significant comment and suggestion. The detailed procedure literature have been given to the synthesis/preparation of Mg(OH)2 NPs. The revised manuscript is shown as below:

(Line 81-91, page 2)

2.2. Synthesis of Mg(OH)2 NPs

The Mg(OH)2 NPs were prepared from the co-precipitation of MgCl2·6H2O, PEG and NH3·H2O at 60 oC for 1.5 h[26-27]. The 200 mL MgCl2 solutions (1.5 mol/L) were added to a 500 mL three-mouth flask, 0.4 g PEG was added as dispersant, and the three-mouth flask was placed in an ultrasonic bath to accelerate the dissolution of PEG. Then 5 wt% NH3·H2O was added slowly dropwise into the mixture (n(OH-)/n(Mg2+)≥2). The mixture was stirred at 300 rpm at 40℃ for 1.5 h. The resulting suspension was cooled to room temperature (20 oC) and aged for 12 h to allow complete precipitation of Mg2+. The resulting suspension was filtered and washed with ionized water and anhydrous ethanol, respectively. It was dried under vacuum at 80 oC for 4 h and ground uniformly to obtain the product nano-Mg(OH)2[26-27].

2.Regarding the NP’s characterization, additional EDS or XPS analysis could provide relevant information regarding the surface of the Mn(OH)2 NPs, and their elemental composition.

The EDS and XPS analysis have been added in our research, which could provide the surface relevant information and elemental composition of the Mn(OH)2 NPs. The revised manuscript is shown as below:

(Line 98-103, page 3)

The elemental composition of the sample was tested by the scanning electron microscope (EDS, X-Max50, Oxford Instruments, UK) and X-ray photoelectron spectroscopy (XPS, Thermo ESCALAB 250Xi, Al K-Alpha) on a VG MultiLab 2000 X-ray photoelectron spectrometer (Thermo Electron Corporation) using Al-Kα (hλ=1486.6 eV) radiation as the excitation source, and the spectra were calibrated by the C 1s peak (284.8 eV).  

(Line 246, page 7)

(Line 177-194, page 5-6)

The composition of the surface elements of the synthesized Mg(OH)2 NPs products was studied by EDS spectrometer, and the EDS representing the strong characteristic peaks of magnesium and oxygen elements were shown. No other impurity elements were observed in the spectrum, indicating the high purity of the nanoparticle structure. The surface scanning of elements C, O, N and Mg is shown in Figure. 2c. The elements are presented with different colors respectively. The atomic percentage ratio of Mg and O elements was 27.41:56.69, and the atomic number ratio of Mg and O elements was close to 1:2. The weight percentages of Mg and O in the product were 42.34 and 52.32, respectively, which were consistent with the theoretical values. The EDS spectra showed that the synthesized product was Mg(OH)2 NPs.

XPS was used to analyze the surface chemical composition of Mg(OH)2 NPs. Figure 2d is the full-scan spectrum of the sample, which shows that the test sample mainly contains the characteristic peaks of Mg, O and C. No miscellaneous peaks of other substances appeared in the Figure. 2d, indicating high purity of the product. The XPS atlas shows that the characteristic peak of Mg2p appears at 49.6 eV and Mg1s at 1303.14eV. In the sample, Mg2p fits only one peak, which should be attributed to Mg2+ in Mg(OH)2. The characteristic peak of O1s appears at 531.12eV, which can also indicate the high purity of the generated Mg(OH)2 NPs.

3.To my view, Table 2 is useless, since all runs and entries have the same (zero) value. A simple comment in the text seems sufficient.

Thank you very much for your valuable advice, the Table 2 can clearly describe the score of the acute eye irritation. Thanks again for your warning, and we will consider your advice in our next work.

Round 2

Reviewer 1 Report

The manuscript describes assessment of Mg(OH)2 nanoparticle (NP) biosafety in several techniques.  The conclusion, that there is no toxic effect of Mg(OH)2 NP under high-dose conditions, is well supported by the results.  Additional information such as how to evaluate in detail was added in the modified manuscript according to the suggestion of 1st version of the manuscript.  The manuscript should be published in JFB without any revision.